# Insecticidal Effect of *Pistacia lentiscus* (Anacardiaceae) Metabolites against *Lobesia botrana* (Lepidoptera: Tortricidae)

Ioanna Dasenaki [1], Petri-Christina Betsi [1], Dimitris Raptopoulos [2] and Maria Konstantopoulou [1,*]

[1] Chemical Ecology and Natural Products Laboratory, Institute of Biosciences and Applications, NCSR "Demokritos", 15341 Athens, Greece; dasenaki@bio.demokritos.gr (I.D.); betsipetri@bio.demokritos.gr (P.-C.B.)

[2] Novagrica Hellas S.A., TESPA "Lefkippos", 15341 Athens, Greece; dimrapto@novagrica.com

\* Correspondence: mkonstan@bio.demokritos.gr

**Abstract:** The extensive use of synthetic insecticides in agriculture poses a great risk for human health and the ecosystem, and mandates the development of safer alternatives derived from natural products. In the present study, we assessed the larvicidal effect of *Pistacia lentiscus* fruits, leaves, and bark extracts and their components on larvae of a major vine pest, *Lobesia botrana*. *Pistacia lentiscus* is an evergreen shrub or small tree possessing significant medicinal value with numerous therapeutic uses since antiquity. Using petri dish residual exposure and topical application bioassays we demonstrated that the fruit extract of *P. lentiscus* and its metabolites were toxic on *L. botrana* larvae. Extracts from leaves and bark showed no effect. Bioassay-guided fractionation of *P. lentiscus* fruit hexane extract led to the identification of its constituents with insecticidal properties on *L. botrana* larvae. Specifically, we have identified that the main contributor to the bioactivity of the hexane extract of *P. lentiscus* fruits is its major fraction, $PLF_{He2}$ (76.25%). Furthermore, we have found that $PLF_{He2}$ is a mixture of triglycerides and that the fatty acids responsible for the observed toxicity are oleic and linoleic acid.

**Keywords:** medicinal plants; grapevine moth; larvicidal activity; fatty acids; petri dish bioassay; topical application

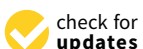

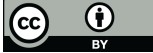

## 1. Introduction

The grapevine moth *Lobesia botrana* (Denis & Schiffermüller), (Lepidoptera: Tortricidae) is one of the most harmful vine pests worldwide. It has major economic impact on the viticulture industry [1], which is among the highest growing agricultural industries globally. With almost 90% of the world's organic grape area grown in Europe its occurrence has a significant economic importance in grapevine-growing areas in the Mediterranean region. Recently it has expanded its distribution range and has been found in Chile, California, and Argentina [2]. *L. botrana* causes damage through inflorescence consumption and berry infestation, which increases susceptibility to pathogens such as *Botrytis cinerea*, thus reducing yield and the quality of wine and grape products [3]. Synthetic insecticides are used extensively to protect vineyards from infestations, but are hazardous to human health and the ecosystem and also pose the risk of insect resistance development [4]. In addition, *L. botrana* populations have developed resistance to many different chemical insecticides.

To mitigate these dangers, alternative, safer approaches are increasingly investigated and incorporated in integrated pest management, such as the use of biological agents (*Bacillus thuringiensis*) or pheromone mediated techniques (mating disruption) [3]. Research has been also focused on the isolation of natural, specialized, and biodegradable insecticides from plants.

In this context, secondary plant metabolites from essential oils of medicinal and aromatic plants may play an important role as population control agents of *L. botrana*.

Mondy et al. [5] found that saw-wort, *Serratula tinctoria* (Asteraceae), extracts, when incorporated in artificial diet, induce significant mortality rates to first, second, and third larval instar, as well as impede larval growth and pupal development. Moreover, tancy, *Tanacetum vulgare* (Asteraceae), essential oil and flowers have been shown to exhibit adult male mortality and affect adult reproductive behavior [6]. Furthermore, essential oils from plant members of the Asteraceae family, such as *Chrysanthemum*, *Tanacetum*, and *Artemisia* genera have been studied for their ability to protect vineyards. When used as cover crops, they have exhibited oviposition deterrence, ovicidal activity, and effective reduction of infestation from *L. botrana*, especially when co-cultivated [7].

Lentisk, *Pistacia lentiscus* L. is an evergreen shrub or small tree belonging in the *Anacardiaciae* family and is widely distributed across the Mediterranean region [8], possessing significant medicinal value with numerous therapeutic uses since antiquity. Extensive research has been focused on the biomedical and pharmacological properties of constituents in resin and aerial parts extracts, most notably on the antioxidant, antimicrobial, anti-inflammatory, wound-healing, anticancer, and hepatoprotective actions [9–16]. The aerial parts are rich in monoterpenes such as limonene, myrcene, α- and β-pinene, terpine-4-ol, and α-terpineol, all compounds with well-established insecticidal activity [17,18]; several studies have investigated the effects of *P. lentiscus* extracts on various insect species, mainly stored products pests [19]. Bachrouch et al. demonstrated the potent fumigant toxicity of the essential oil from leaves against adults and larvae of red flour beetle *Tribolium castaneum* (Coleoptera: Tenebrionidae) [20] and adults of cigarette beetle *Lasioderma serricorne* (Coleoptera: Ptinidae) [21]. In another study, besides adult fumigant toxicity, the group found that essential oil from *P. lentiscus* leaves decreased longevity and copulation, fecundity, and hatching rates of the lepidopterans carob moth *Ectomyelois ceratoniae* (Lepidoptera: Pyralidae) and Mediterranean flour moth *Ephestia kuehniella* (Lepidoptera: Pyralidae) [22]. Strong ovicidal activity of *P. lentiscus* essential oil was reported against the Hessian fly, *Mayetiola destructor* (Diptera: Cecidomyiidae) [23], whilst non-polar extracts of branch and leaves exhibited effective repellency when incorporated in artificial diet of *T. castaneum* [24].

These reports highlight the promising prospect of *P. lentiscus* as a source of a natural and safe population control agent of *L. botrana* in order to protect vineyards. In view of this potential our main goal was to assess the larvicidal effect of *P. lentiscus* fruits, leaves, and bark extracts and its components on *L. botrana*.

## 2. Materials and Methods

### 2.1. Plant Material

Fruits from *Pistacia lentiscus* were collected from Sygrou Park, Athens (38°03′50.1″ N 23°48′52.7″ E) in 2018. The plant material was separated into leaves, fruits, and bark (the latter was cut into 1 cm pieces), and 100 g of each plant part was washed with HPLC-grade water and dried at room temperature for 3 h.

### 2.2. Extraction and Fractionation

Crude hydromethanolic extracts of *P. lentiscus* fruits, leaves, and bark were prepared by maceration of 100 g material in 800 mL 80% MeOH (80:20 HPLC grade MeOH:H$_2$O) for 24 h. First, 50 g of material was added to 100 mL 80% MeOH and was homogenized using an Omni Mixer (Sorvall, Kennesaw, GA, USA). The homogenate was transferred to a 1 L-volume conical flask and 300 mL of MeOH 80% was added (total solvent volume in flask: 400 mL). The homogenization was repeated with another 50 g of material and transferred to a separate conical flask (duplicate). Bark material was not homogenized. The samples were sonicated for 1 h and then left under continuous stirring at room temperature for 24 h. The following day, the samples were vacuum-filtered, and the filtrate was collected and concentrated under reduced pressure using a rotary evaporator (crude hydromethanolic yields: 11.1%, 26.22%, and 7.13% for fruits (PLF$_{Me}$), leaves (PLL$_{Me}$), and bark (PLB$_{Me}$) of the initial material, respectively).

The solid residue of the filtration was collected, left to dry at room temperature in a fume hood for 24 h, and subjected to further maceration in 500 mL n-hexane (HPLC grade) at room temperature for 72 h under continuous stirring. The hexane extract was collected by vacuum-filtration and the filtrate was concentrated under reduced pressure in a rotary evaporator, producing a yellowish-green oil (yield: 13.84%) (PLF$_{He}$). The active extract (fruits) PLF$_{He}$ was analyzed and its constituents were tentatively identified on a GC-MS. GC-MS analyses were conducted using an Agilent Technologies (Agilent Technologies Inc., Santa Clara, CA, USA) 7820A gas chromatograph equipped with a HP-5MS capillary column (30 m × 0.25 mm, film thickness 0.25 μm), an Agilent Technologies 5977B MS detector operating in electron ionization mode at 70 eV and an Agilent Technologies 7693A automatic liquid sampler. Injection was performed at 220 °C in a split ratio 1:5, the ion source temperature and transfer line temperature were set at 230 and 250 °C, respectively, the carrier gas was He at 1.4 mL min$^{-1}$ and the oven temperature was increased from 60 to 300 °C at a rate of 3 °C min$^{-1}$ and subsequently held at 300 °C for 10 min.

The crude hydromethanolic extract was further fractionated using solvents of increasing polarity (Figure 1). First, 5 g of each extract was resuspended in 100 mL 80% MeOH and subjected to liquid–liquid extraction with 100 mL petroleum ether (analytical grade). The extraction was repeated three times and the collected layers were concentrated in a rotary evaporator (petroleum ether fractions yields: 4.92%, 4.80%, and 5.80% for fruits (PLF$_{PE}$), leaves (PLL$_{PE}$), and bark (PLB$_{PE}$), respectively).

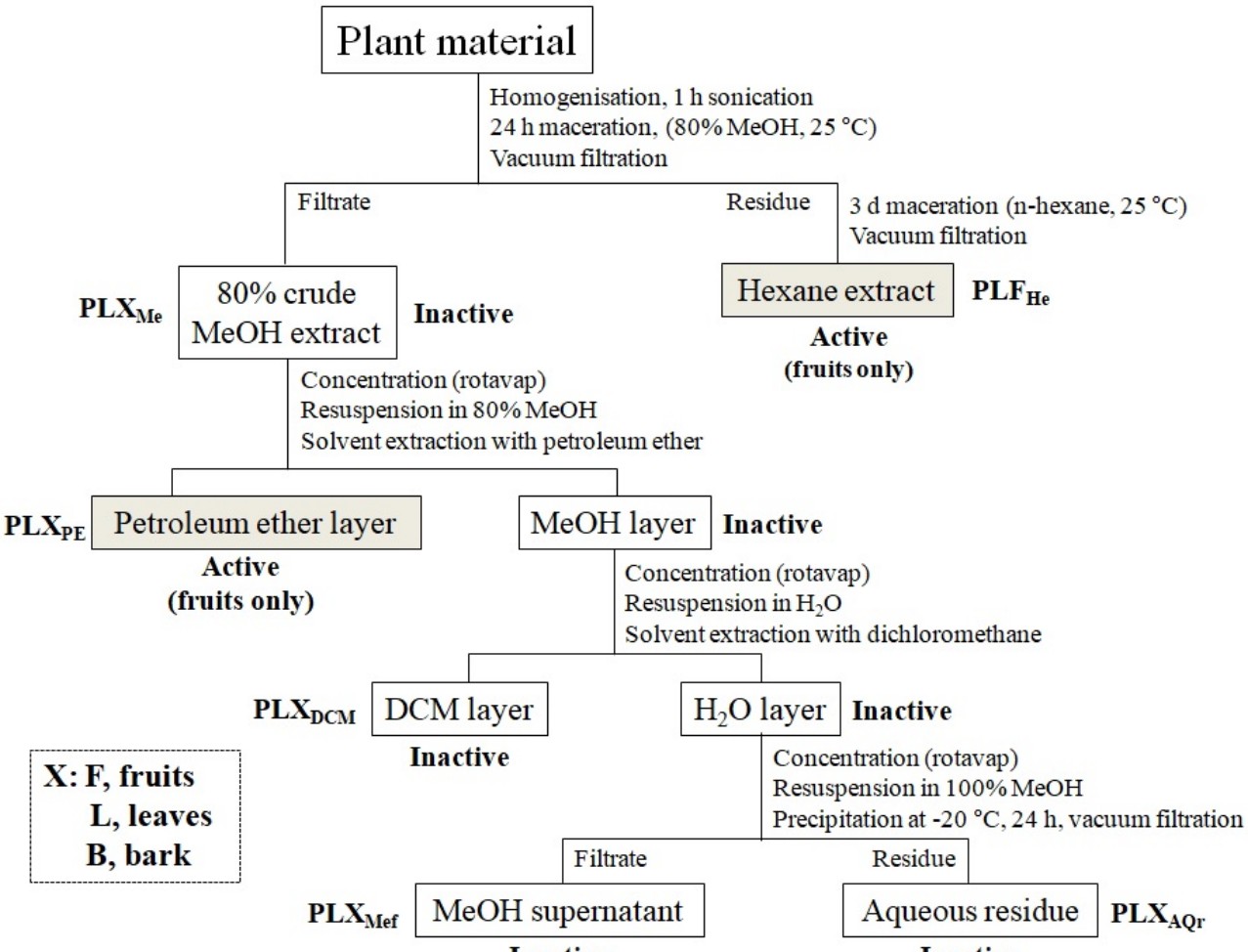

**Figure 1.** Overview of the initial purification scheme of fruits (PLF), of *P. lentiscus*, and the respective insecticidal activity of each fraction on *L. botrana* larvae.

The hydromethanolic layer was evaporated to dryness (rotavap), resuspended in HPLC-grade water, and extracted with dichloromethane (DCM, HPLC-grade). The extraction was repeated three times and the DCM layers were combined and concentrated under reduced pressure using a rotary evaporator (DCM fractions yields: 3.20%, 4.00%, and 3.80% for fruits (PLF$_{DCM}$), leaves (PLL$_{DCM}$), and bark (PLB$_{DCM}$), respectively). The aqueous layer was dried, resuspended in 100% MeOH, and placed at $-20\,^{\circ}$C for 24 h, allowing for proteins and other polar compounds to precipitate. The supernatant methanol solution was vacuum-filtered at 4 $^{\circ}$C and the filtrate was collected and dried under reduced pressure with a rotary evaporator (52.60%, 44.20%, and 42.20% yield for fruits (PLF$_{Mef}$), leaves (PLL$_{Mef}$), and bark (PLB$_{Mef}$), respectively). The precipitate and filtration residue were resuspended in HPLC grade water and concentrated to dryness, resulting in 27.80%, 6.60%, and 24.20% yield for fruits (PLF$_{AQr}$), leaves (PLL$_{AQr}$), and bark (PLB$_{AQr}$), respectively. All extracts were resuspended in their respective solvents, transferred to glass vials, and stored at $-20\,^{\circ}$C until use.

A portion of the fruits' hexane extract (2 g) was chromatographed on a 2.5 cm diameter silica gel (Kieselgel 60, Merck, Darmstadt, Germany) gravity column, using a n-hexane/ethyl acetate step gradient (100, 95:5, 90:10, 85:15, 80:20, 70:30, 60:40, 50:50, and 0:100). Two fractions of 100 mL each were collected from each solvent system, resulting in 18 fractions in total. All fractions were evaluated by thin layer chromatography (Kieselgel 60 F254 aluminum plates), developed with a hexane and ethyl acetate 70:30 mobile phase, using a 15% H$_2$SO$_4$ in MeOH reagent and brief heating. Fractions with similar chromatographic characteristics were combined to give five final fractions: 1–4: PLF$_{He1}$, 5: PLF$_{He2}$, 6–8: PLF$_{He3}$, 9: PLF$_{He4}$, and 10–18: PLF$_{He5}$ (Figure 2).

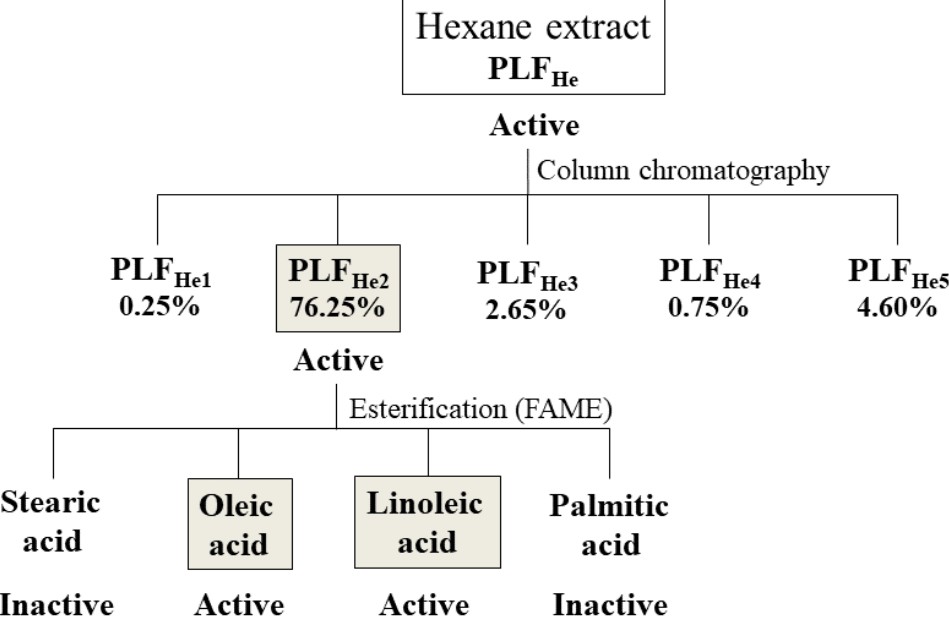

**Figure 2.** Overview of the purification process of the hexane extract of *P. lentiscus* fruits (PLFHe) and the respective insecticidal activity of each fraction or constituent on *L. botrana* larvae.

### 2.3. Triglyceride Identification

Fraction PLF$_{He2}$, which was found to exhibit strong bioactivity in preliminary bioassays with *L. botrana* larvae, was further characterized by $^1$H NMR (Bruker DRX 400, Billerica, MA, USA). Based on the resulting spectrum, which revealed distinctive chemical shifts, PLF$_{He2}$ was identified as a mixture of triglycerides (Figure S1 in Supplementary Materials).

Briefly, 10 mg of PLF$_{He2}$ was weighed in a 1.5 mL Eppendorf tube, dissolved in 1.5 mL of hexane, and, after adding 200 µL of KOH/MeOH 2 M, the mixture was homogenized by moderate vortexing. Finally, 0.4 g Na$_2$SO$_4$ was added and the sample was vortexed again. The Eppendorf tube was placed at $-20\,^{\circ}$C for 30 min, then 200 µL of the up-

per layer containing the FAMEs was recovered and resuspended in 1.8 mL of hexane. The fatty acids comprising the triglycerides were converted to fatty acid methyl esters (FAMEs) for further analysis by GC-MS. With the use of Supelco 37 component FAME mix standard (Sigma-Aldrich, St. Louis, MO, USA), the fatty acids were identified predominantly as oleic acid, linoleic acid, palmitic acid, and, to a lesser extent, stearic acid (Figures S2–S6 in Supplementary Materials) [25].

The resulting FAMEs were analyzed by GC-MS. GC-MS analyses were conducted using an Agilent Technologies 7820A gas chromatograph equipped with a HP-5MS capillary column (30 m × 0.25 mm, film thickness 0.25 μm), an Agilent Technologies 5977B MS detector operating in electron ionization mode at 70 eV, and an Agilent Technologies 7693A automatic liquid sampler. Injection was performed at 220 °C in a split ratio 1:5, the ion source temperature and transfer line temperature were set at 230 and 250 °C, respectively, the carrier gas was He at 1.4 mL min$^{-1}$, and the oven temperature was increased from 60 to 300 °C at a rate of 3 °C min$^{-1}$ and subsequently held at 300 for 10 min.

### 2.4. Insects

A laboratory colony of the European grapevine moth *Lobesia botrana* from feral populations from Northern Greece was established at the Chemical Ecology and Natural Products Laboratory of NCSR «Demokritos». Larvae were reared on artificial diet. All life stages were kept at a 16:8 (L:D) photoperiod, at 22 ± 1 °C and 60–70% humidity. The weight of the larvae used for the bioassays was 12 ± 0.7 mg.

### 2.5. Petri Dish Residual Exposure Bioassay

Glass petri dishes of the following dimensions: bottom internal diameter, 5 cm; rim height, 1.3 cm; and lid internal diameter, 5.7 cm, with a total surface area 65.5 cm$^2$, were used for residual exposure bioassays on *L. botrana* larvae. Prior to each bioassay, petri dishes were cleaned by 2-h sonication in water (2% detergent), copious rinsing with deionized water, then acetone, and finally dried at 70 °C for 2 h. Each *P. lentiscus* fraction was diluted in its respective solvent to produce a series of solutions of different concentrations. All of the aforementioned fractions were tested for their bioactivity against 5th larval instar. With the aid of a micropipette, 299, 312, and 389 μL of each solution was applied to the bottom, rim, and lid, respectively, so that all treated surfaces were covered with the same sample concentration (micrograms per square centimeter). After application, the petri dishes were rotated manually under a fume hood until solvent evaporation to achieve an even distribution of the sample. Petri dishes were left uncovered in a fume hood for 1 h to ensure complete evaporation of solvent traces [26,27]. The concentrations tested ranged from 5 mg/mL (76 μg/cm$^2$) to 25 mg/mL (382 μg/cm$^2$). Ten 5th larval instar were placed in each petri dish. Two control treatments were used in each experiment consisting of (a) a clean, untreated petri dish (control), and (b) a petri dish treated with solvent only (solvent control) and were run simultaneously with the sample treatments. Three replicates were used for each treatment. Exposure time was 3 h. At the end of the exposure, the larvae were transferred to flat-bottomed, lidded, 24-well polystyrene plates and were provided with solid larval diet cubes. Insect mortality was recorded at 3, 24, 48, and 72 h.

The hexane fraction of *P. lentiscus* fruits (PLF$_{He}$) exhibited the highest mortality rates and, therefore, its subfractions (PLF$_{He1}$-PLF$_{He5}$) were tested for bioactivity. Of all the subfractions tested, PLF$_{He2}$ exhibited strong insecticidal activity at 382 μg/cm$^2$. Further analysis revealed that PLF$_{He2}$ is a mixture of triglycerides, comprising of oleic acid, linoleic acid, stearic acid, and palmitic acid; henceforth, these fatty acids were also tested for their bioactivity at concentrations ranging between 5 mg/mL and 15 mg/mL.

### 2.6. Topical Application Bioassay

The hexane extract of *P. lentiscus* fruits (PLF$_{He}$) and its PLF$_{He2}$ fraction were diluted in acetone to give solutions at concentrations ranging from 10 to 100 mg/mL for PLF$_{He}$ and 75 mg/mL for PLF$_{He2}$. Fifth larval instar *L. botrana* were placed in a Petri dish

(diameter 5 cm) and aliquots of 2 μL/insect of each solution (resulting in 20, 40, 80, 160, and 200 μg doses for PLF$_{He}$ and 150 μg for PLF$_{He2}$), were dorsally applied on larvae using a micropipette. After one minute, to allow for solvent evaporation, the larvae were transferred to a lidded 24-well plate, each larva placed in an individual well with a cube of artificial diet. The wells were kept at 24 ± 1 °C and 60–70% RH. Larvae mortality was recorded after 24, 48, and 72 h. Control treatments consisted of untreated larvae and acetone treatment only. Ten larvae were used for each treatment and seven replicates were conducted.

*2.7. Statistical Methods*

Data were subjected to analysis of variance (ANOVA) (SAS Institute, 2000, Cary, NC, USA). The means of data were separated using the Duncan's multiple range tests (MRT) at $p < 0.05$. Data obtained from each concentration or dose of larvicidal bioassay were subjected to Probit analysis; LC$_{50}$ values and slopes were calculated (IBM SPSS vs. 22).

**3. Results**

The crude hydromethanolic extract of fruits, leaves, and bark of *P. lentiscus* did not exhibit insecticidal activity on *L. botrana* larvae using Petri dish residual exposure bioassays. Specifically, the exhibited mortality was zero in all cases. Of the sequentially resulting fractions from each solvent, only the petroleum ether and hexane fraction from fruits demonstrated larvicidal activity, suggesting the non-polar nature of the active compounds. The hexane fraction, PLF$_{He}$, was more potent than the petroleum ether one, PLF$_{PE}$ (data not shown). In addition, the PLF$_{PE}$ yield was considerably lower than that of PLF$_{He}$; thus, the latter was selected for further bioassays.

GC-MS results showed that the major constituents of the PLF$_{He}$ extracts was comprising mainly of aliphatic acids (palmitic acid 12.3%, stearic acid 2%, and linoleic and oleic acids 25%) and their respective methyl esters (methyl palmitate 5.8%, methyl linoleate 3.7%, and methyl oleate 5%) making up for 54.1% of the total. The remaining identified compounds were mainly terpenes and sesquiterpenes at 8.8%, phenolic compounds 28.4%, and γ-sitosterol 4.7% (Table S1 in Supplementary Materials).

Sitosterol was tentatively identified through GC-MS in fraction PLF$_{He4}$ which yielded a residue of 15 mg as white powder (Figure S7 in Supplementary Materials).

In comparison, 90.64% of the mildly active PLF$_{Pe}$ was comprised of phenolic compounds with the remaining methylated acids at 7.98% and the inactive PLL$_{Pe}$ (leaf extract) consisted up to 80.62% of methylated fatty acids.

Petri dish residual exposure bioassays revealed that PLF$_{He}$ had a significant insecticidal activity on *L. botrana* larvae and the mortality was concentration dependent (3 h: F = 28.307, df = 5, $p \leq 0.001$; 24 h: F = 46.302, df = 5, $p \leq 0.001$). The larval mortality ranged from 4% (5 mg/mL) to 63% (25 mg/mL) after 24 h and 28.5% after 3 h (Table 1). Probit analysis of the concentration mortality response revealed the LC$_{50}$ value of 287.85 μg/cm$^3$ at 24 h was significantly different than at the 3 h (441.2 μg/cm$^3$) (Table 2). Control mortality was zero.

**Table 1.** Larvicidal activity (%) of PLF$_{He}$ to *L. botrana* recorded at 3 and 24 h after Petri dish residual exposure bioassay. Means followed by the same letter within each column are not significantly different (Duncan's multiple range tests [MRT] test $p > 0.005$).

| Concentration (μg/cm$^3$) | Time (h) | |
|---|---|---|
| | 3 | 24 |
| 76 | 0 ± 0 a | 4 ± 1.5 a |
| 115 | 1.5 ± 0.8 a | 15.5 ± 3.3 b |
| 153 | 1 ± 0.8 a | 25 ± 3.2 b |
| 229 | 12 ± 3.5 b | 48.5 ± 4.2 c |
| 305 | 25 ± 3.5 c | 55.5 ± 4.1 d |
| 382 | 28.5 ± 2.9 c | 63 ± 4 d |

**Table 2.** Larvicidal activity of $PLF_{He}$ recorded at 3 and 24 h after Petri dish residual exposure bioassay. $LC_{50}$ values are considered significantly different when 95% CL fail to overlap. Since goodness-of-fit test is significant ($p < 0.05$), a heterogeneity factor is used in the calculation of confidence limits (CL).

| Time (h) | $LC_{50}$ ($\mu g/cm^3$) | CL 95% | Slope $\pm$ SE | Intercept $\pm$ SE | $x^2$ | $p$ |
|---|---|---|---|---|---|---|
| 3 | 441.2 | 410.7–484.2 | $0.007 \pm 0.001$ | $-2.87 \pm 0.05$ | 1448.07 | 0.000 |
| 24 | 287.85 | 269.9–308.9 | $0.006 \pm 0.003$ | $-1.66 \pm 0.03$ | 1745.48 | 0.000 |

The effect of $PLF_{He}$ in topical application was also dose-dependent (3 h: F = 12.559, df = 3, $p \leq 0.001$; 24 h: F = 10.091, df = 3, $p \leq 0.001$; 48 h: F = 10.109, df = 3, $p \leq 0.001$; 72 h: F = 8.910, df = 3, $p \leq 0.001$). Larval mortality ranged from 22.7% (20 $\mu g$/insect) to 54% (200 $\mu g$/insect) after 72 h (Table 3), displaying significant $LD_{50}$ value on topical application that ranged from 148.9 to 239.9 $\mu g$/insect after 3 to 72 h, respectively (Table 4).

**Table 3.** Larvicidal activity (%) of $PLF_{He}$ to *L. botrana* recorded at 3 and 24 h after topical bioassay. Means followed by the same letter within each column are not significant different (Duncan's multiple range tests [MRT] test $p > 0.005$).

| Dose (µg/Insect) | Time (h) | | | |
|---|---|---|---|---|
| | 3 | 24 | 48 | 72 |
| 20 | $1.7 \pm 1.7$ a | $6.7 \pm 3.8$ a | $16.7 \pm 4.8$ a | $22.7 \pm 5.4$ a |
| 40 | $12 \pm 4.9$ b | $20 \pm 6.3$ b | $22 \pm 4.7$ a | $22.6 \pm 5.2$ a |
| 80 | $32 \pm 5.8$ c | $40 \pm 4.57$ c | $44 \pm 5.1$ b | $44 \pm 5.1$ b |
| 160 | $28 \pm 4$ c | $30 \pm 6.3$ b | $52 \pm 4.7$ c | $58 \pm 6.6$ c |
| 200 | $40 \pm 5.5$ d | $48 \pm 5.8$ d | $50 \pm 6.3$ c | $54 \pm 6$ c |

**Table 4.** Larvicidal activity of $PLF_{He}$ recorded at 3, 24, 48, and 72 h after topical application bioassay. $LD_{50}$ values are considered significantly different when 95% CL fail to overlap. Since goodness-of-fit test is significant ($p < 0.05$), a heterogeneity factor is used in the calculation of confidence limits (CL).

| Time (h) | $LD_{50}$ ($\mu g$) | CL 95% | Slope $\pm$ SE | Intercept $\pm$ SE | $x^2$ | $p$ |
|---|---|---|---|---|---|---|
| 3 | 239.9 | 189.8–372.8 | $0.005 \pm 0.000$ | $-1.41 \pm 0.06$ | 258.76 | 0.000 |
| 24 | 217.6 | 167.7–364.0 | $0.005 \pm 0.000$ | $-1.07 \pm 0.05$ | 258.96 | 0.000 |
| 48 | 168.6 | 135.2–232.8 | $0.005 \pm 0.000$ | $-0.852 \pm 0.048$ | 188.934 | 0.000 |
| 72 | 148.9 | 118.1–200.8 | $0.005 \pm 0.000$ | $-0.765 \pm 0.47$ | 192.07 | 0.000 |

Considering its larvicidal activity, $PLF_{He}$ was further fractionated by column chromatography, resulting in five final subfractions ($PLF_{He1}$–$PLF_{He5}$), which were tested for their bioactivity through Petri dish residual bioassays at their percent concentration compared to $PLF_{He}$ at 382 $\mu g/cm^3$ (Figure 3). Subfraction $PLF_{He2}$, gave an oily yellow compound (1.43 g), which accounted for the majority of the initial hexane extract, with a yield of 76.25%. $PLF_{He2}$, which accounted for the majority of the initial $PLF_{He}$ extract (76.25%), was the only one to exhibit considerable mortality against *L. botrana* larvae (3 h: F = 5.420, df = 4, $p = 0.001$; 24 h: F = 26.480, df = 4, $p \leq 0.001$; 48 h: F = 31.055, df = 4, $p \leq 0.001$; 72 h: F = 43.319, df = 4, $p \leq 0.001$). Even at 382 $\mu g/cm^3$, $PLF_{He2}$ induced larval mortality, which reached 77.5% within 72 h using Petri dish residual assays (F = 5.134, df = 3, $p = 0.009$), whilst topical bioassays resulted in a 50% mortality at 150 $\mu g$/insect in 72 h (Table 5) (F = 21.887, df = 3, $p \leq 0.001$).

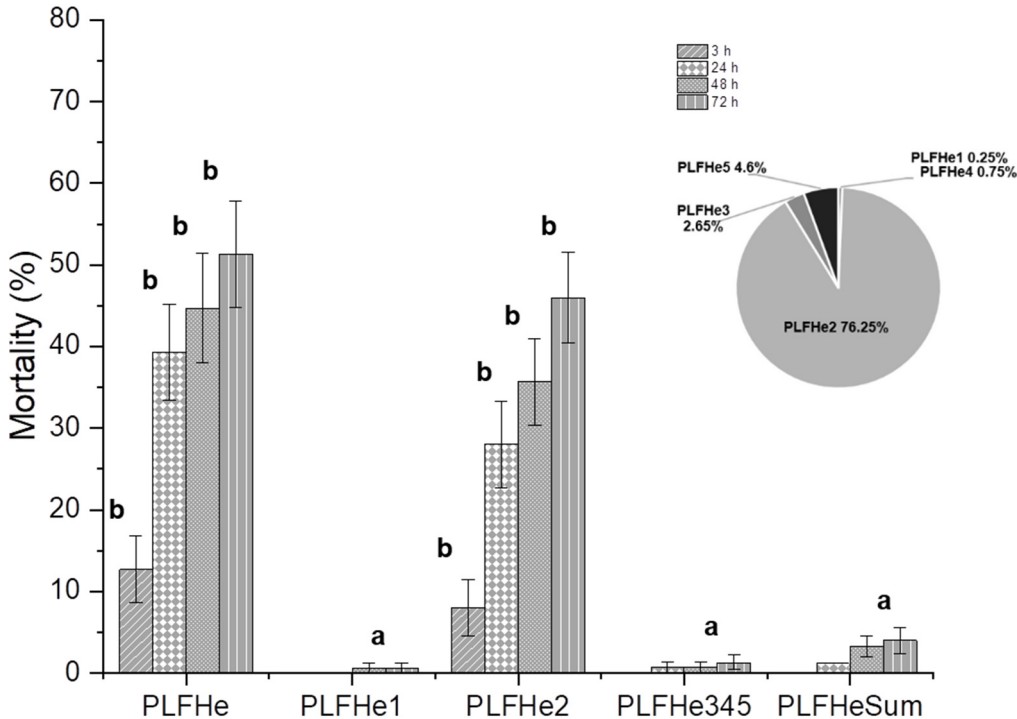

**Figure 3.** Larvicidal activity (%) of *L. botrana* to $PLF_{He}$ and its fractions ($PLF_{He}$ at 25 mg/mL and percentage of contribution of each fraction to the $PLF_{He}$, pie graph) recorded at 3, 24, 48, and 72 h after Petri dish residual exposure bioassay. Means followed by the same letter within each column (fraction/h) are not significantly different (Duncan's multiple range tests [MRT] test *p* > 0.005).

**Table 5.** Larvicidal activity (%) of $PLF_{He2}$ to *L. botrana* recorded at 3, 24, 48, and 72 h after Petri dish residual exposure bioassay at 25 mg/mL (first row) and topical bioassay at 75 µg dose (second row). Means followed by the same letter within each row are not significantly different (Duncan's multiple range tests [MRT] test *p* > 0.005).

| | Time (h) | | | |
|---|---|---|---|---|
| **Concentration (µg/cm$^3$)** | **3** | **24** | **48** | **72** |
| 382 | 18.3 ± 5.3 a | 60.8 ± 5.9 b | 71.7 ± 5.7 c | 77.5 ± 5.86 c |
| **Dose (µg/Insect)** | | | | |
| 150 | 25 ± 3.4 a | 31.7 ± 4 ab | 38.3 ± 5.3 b | 50 ± 5.8 c |

The observed bioactivity of $PLF_{He2}$ in both types of assays, suggesting that $PLF_{He2}$ comprises of the majority of the compounds responsible for the bioactivity observed in $PLF_{He}$.

Based on its $^1$H NMR signal, $PLF_{He2}$ was identified as a mixture of triglycerides (Figure S1 in Supplementary Materials). The fatty acids comprising the triglycerides were converted to fatty acid methyl esters (FAMEs) for further analysis by GC-MS. The fatty acids identified were predominantly oleic acid, linoleic acid, palmitic acid, and, to a lesser extent, stearic acid (Figures S2–S6 in Supplementary Materials). Using standard compounds, all four fatty acids were tested for their bioactivity by Petri dish residual exposure bioassay, from 76 µg/cm$^3$ to 229 µg/cm$^3$ concentration range. Whilst stearic and palmitic acid showed no activity (zero mortality), oleic and linoleic acid demonstrated strong larvicidal activity in a dose-dependent fashion (Table 6). The mortality in the oleic acid reached 90.7% even after 3 h, and 98.7% after 72 h (3 h: F = 32.564, df = 4, *p* ≤ 0.001; 24 h: F = 29.162, df = 4, *p* ≤ 0.001; 48 h: F = 28.157, df = 4, *p* ≤ 0.001; 72 h: F = 35.254, df = 4, *p* ≤ 0.000), and in the linoleic acid, 79.3% and 83.3% after 3 and 72 h, respectively (3 h: F = 44.922, df = 4, *p* ≤ 0.001; 24 h: F = 43.550, df = 4, *p* ≤ 0.001; 48 h: F = 46.395, df = 4, *p* ≤ 0.001; 72 h: F = 49.529, df = 4,

$p \leq 0.001$). The LD$_{50}$ for oleic acid ranged from 172.33 to 112.89 μg/cm$^3$ and for linoleic acid from 201.48 to 157.26 μg/cm$^3$. The LD$_{50}$ values of both fatty acids (Table 7) decreased significantly compared to that of the initial hexane extract.

**Table 6.** Larvicidal activity (%) of oleic acid and linoleic acid to *L. botrana* recorded at 3, 24, 48, and 72 h after Petri dish residual exposure bioassay. Means followed by the same letter within each column are not significantly different (Duncan's multiple range tests [MRT] test $p > 0.005$).

| Concentration (μg/cm$^3$) | Time (h) | | | |
|---|---|---|---|---|
| | **3** | **24** | **48** | **72** |
| | **Oleic Acid** | | | |
| 76 | 10.7 ± 3.1 a | 20 ± 5.5 a | 23.3 ± 5.9 a | 24.7 ± 5.7 a |
| 116 | 32.7 ± 7.3 b | 51.3 ± 6.2 b | 57.3 ± 6.5 b | 60.7 ± 6.2 b |
| 153 | 56 ± 5.9 c | 68 ± 5.2 c | 69.3 ± 4.9 c | 71.3 ± 4.6 b |
| 191 | 64.7 ± 6.1 c | 75.3 ± 5.1 c | 77.3 ± 4.7 c | 84 ± 4.2 c |
| 229 | 90.7 ± 2.8 d | 92.7 ± 2.8 c d | 93.3 ± 2.7 d | 98.7 ± 0.9 d |
| | **Linoleic Acid** | | | |
| 76 | 6 ± 2.9 a | 14.6 ± 3.9 a | 19.3 ± 3.8 a | 20.7 ± 4.1 a |
| 116 | 13.3 ± 3.2 b | 28 ± 2.9 b | 32 ± 3.2 b | 32.7 ± 3.4 b |
| 153 | 44 ± 4.4 c | 57.3 ± 3.6 c | 60.7 ± 3.1 c | 62 ± 3.1 c |
| 191 | 65.3 ± 7.0 d | 71.3 ± 5.7 d | 74.7 ± 5.0 d | 76.7 ± 4.6 d |
| 229 | 79.3 ± 5.0 d | 80.7 ± 4.7 d | 82 ± 4.3 d | 83.3 ± 3.9 d |

**Table 7.** Larvicidal activity of oleic and linoleic acid recorded at 3, 24, 48, and 72 h after Petri dish residual exposure bioassay. LD$_{50}$ values are considered significantly different when 95% CL fail to overlap. Since goodness-of-fit test is significant ($p < 0.05$), a heterogeneity factor is used in the calculation of confidence limits (CL).

| Time (h) | LD$_{50}$ (μg/cm$^3$) | CL 95% | Slope ± SE | Intercept ± SE | $x^2$ | $p$ |
|---|---|---|---|---|---|---|
| | | **Oleic Acid** | | | | |
| 3 | 172.33 | 152.74–191.05 | 0.010 ± 0.001 | −1.65 ± 0.04 | 2320.14 | 0.000 |
| 24 | 135.37 | 110.15–154.79 | 0.009 ± 0.001 | −1.11 ± 0.04 | 2056.20 | 0.000 |
| 48 | 123.48 | 93.41–144.86 | 0.008 ± 0.001 | −1.01 ± 0.04 | 2073.85 | 0.000 |
| 72 | 112.89 | 81.51–134.61 | 0.011 ± 0.001 | −1.00 ± 0.04 | 2238.15 | 0.000 |
| | | **Linoleic Acid** | | | | |
| 3 | 201.42 | 184.75–220.40 | 0.010 ± 0.001 | −2.08 ± 0.05 | 2169.08 | 0.000 |
| 24 | 173.67 | 141.82–189.52 | 0.009 ± 0.001 | −1.54 ± 0.04 | 1480.62 | 0.000 |
| 48 | 161.81 | 145.05–177.17 | 0.006 ± 0.001 | −1.36 ± 0.04 | 1304.06 | 0.000 |
| 72 | 157.26 | 140.23–172.51 | 0.008 ± 0.001 | −1.34 ± 0.04 | 1302.64 | 0.000 |

## 4. Discussion

The current insect pest control is accomplished by spraying chemicals. Over 98% of sprayed insecticides reach a destination other than their target species, because they are sprayed or spread across entire agricultural fields. According to the Stockholm Convention on Persistent Organic Pollutants, nine out of the twelve most dangerous and persistent chemicals are pesticides. While insecticides have a serious impact on the environment affecting non-target species, including humans, insects are rapidly developing resistance to them. To mitigate the negative environmental and human health impact as consequences of the current practices, it is necessary to make the best use of nature-based innovations to meet the European Green Deal goals to reduce the overall use and risk of chemical pesticides by 50% and the use of more hazardous pesticides by 50% by 2030. Insecticides are not allowed in organic farming, while consumer demand for organic produce is rising rapidly.

Agricultural pest management botanical insecticides are best suited for use in organic food production, as well as in the production and postharvest protection of food [28,29].

*L. botrana* larvae typically develop on inflorescences, unripe grapes, and ripening-ripe grapes and, thus, are exposed to environmental factors. That is why we have tested its resistance to *P. lentiscus* extracts using contact and topical bioassays. Chrysargyris et al. [30] revealed that *M. spicata* essential oil had larvicidal activity on *L. botrana*, displaying a significant $LD_{50}$ value on topical application.

The results obtained in the present study demonstrate that fruit extract of *P. lentiscus* and the metabolites contained in it were toxic to *L. botrana* larvae. Extracts from leaves and bark have showed no effect. Bioassay-guided fractionation of *P. lentiscus* fruit hexane extract led to the separation and identification of fatty acids; oleic acid and linoleic acid were found to have insecticidal properties on *L. botrana* larvae.

$PLF_{He}$ illustrates the increased potency of each compound as an insecticidal agent. In some cases, compounds that exhibit strong bioactivity individually, may be less effective when combined as a mixture [31].

In our work, we have demonstrated that the non-polar compounds of *P. lentiscus* fruits exhibit significant larvicidal activity against *L. botrana*, thus revealing the promising potential of *P. lentiscus* fruits as a source of natural insecticides in order to protect vineyards. Specifically, we have identified that the main contributor to the bioactivity of the hexane extract of *P. lentiscus* fruits is its major fraction, $PLF_{He2}$ (76.25%). Furthermore, we have found that $PLF_{He2}$ is a mixture of triglycerides and that the fatty acids responsible for the observed toxicity are oleic and linoleic acid. Further studies are necessary to identify how these fatty acids (stearic and palmitic acid including) are combined in the triglyceride(s) and the effect they exert on the overall toxicity.

The effect of $PLF_{He}$ was moderate (63%), but promising after 24 h ($LD_{50}$ 287.85 µg/cm$^3$); the effect of $PLF_{He2}$ increased to 77.5% after 72 h; and, finally, the oleic acid and linoleic acid outperformed both, giving mortality of 92.7% and 80.7%, respectively, with significantly lower $LD_{50}$ of 135.37 µg/cm$^3$ and 173.67 µg/cm$^3$ after only 24 h.

To our knowledge, this is the first reference on the insecticidal effect of *P. lentiscus* fatty acids (oleic and linoleic acids). There are studies demonstrating the insecticidal properties of oleic acid and linoleic acid isolated from plants against other insects. Fatty acid methyl esters were proven to be the major constituents of the oil derived from the fruits of *Melia azedarach* (Meliaceae) and may be the main responsible factor for the insecticidal and repellent properties against *Spodoptera littoralis* (Lepidoptera: Noctuidae) [32]. Moreover, linoleic acids derived from *Ricinus communis* (Euphorbiaceae) had insecticidal activities against *Spodoptera frugiperda* larvae (Lepidoptera: Noctuidae) [33].

Fatty acids, such as oleic, linoleic acid, palmitic, and stearic, have been referenced as compounds of *P. lentiscus*, with oleic acid as the major fatty acid in the seed oil [34,35]. Belyagoubi-Benhammou et al. [36], investigating the chemical composition of *P. lentiscus* fruit fatty oil, mentioned oleic, palmitic, linoleic, and stearic acids as the main fatty acids. Oleic acid and linoleic acid were found having insecticidal activity against fourth larval instar *Aedes aegyptii* (Diptera: Culicidae) and exhibited potent feeding deterrent properties against neonate larvae of *Helicoverpa zea* (Lepidoptera: Noctuidae), *Lymantria dispar* (Lepidoptera: Erebidae), *Orgyia leucostigma* (Lepidoptera: Erebidae), and *Malacosoma disstria* (Lepidoptera: Lasiocampidae) [37]. The mosquitocidal assay showed that both oleic and linoleic acids had an $LD_{50}$ value of 100 µg/mL against *A. aegyptii* larvae at 24 h.

In addition, oleic and linoleic acids isolated from *Citrullus colocynthis* (Cucurbitaceae) and *Millettia pinnata* (Fabaceae) were quite potent against fourth larval instar of *A. aegypti* ($LC_{50}$ 8.80, 18.20 and $LC_{90}$ 35.39, 96.33 ppm), *Anopheles stephensi* (Diptera: Culicidae) ($LC_{50}$ 9.79, 11.49 and $LC_{90}$ 37.42, 47.35 ppm), and *Culex quinquefasciatus* (Diptera: Culicidae) ($LC_{50}$ 7.66, 27.24 and $LC_{90}$ 30.71, 70.38 ppm) [38,39].

Fatty acids typically serve as solvents that, in conjunction with emulgators, stabilize the active principles (such as azadirachtin or pyrethrins) in commercial biopesticides [40]. However, recently, conjugated linoleic acid has been characterized as a novel insecticide

targeting the agricultural pest *Leptinotarsa decemlineata* (Coleoptera: Chrysomelidae), which is a major pest of solanaceous crops in USA [41].

The difference on the activity of *P. lentriscus* extracts between the two types of bioassays is reasonable, due to the exposure time, the dose, and the behavior of the larvae. In Petri dish bioassays larvae are forced into contact with the extract constantly for 3 h, while in topical application bioassays, they are free to resume their activity after the application. When designing a formulation for field application, results of both types of laboratory bioassays should be taken into account, along with the typical behavior of a given pest in its ecological niche, in order to result in an effective formulate.

Fatty acids act on the insect' nervous system. They need to go through the cuticle, blood barrier, and perineurium of the insect. They cause death of the basic unit of the nervous system, then disturb the insects' behavior, movement, etc., and ultimately lead the larval poisoning/death [42]. de Melo et al., reported that fatty acids demonstrated insecticidal activity against *C. quinquefasciatus*, and histological analysis of oleic and linoleic acids showed that they could induce cell instability in the midgut cells [43]. However, the medium chain FA (carbon atoms: 7–12) have showed insecticidal activity against *A. gambiae* by blocking the voltage-gated potassium channels (Kv2) of nervous cells [44].

Further work is needed to incorporate oleic and linoleic acids into an effective field insecticide and proper formulation for field application. Finally, it is important to investigate the mode of action of *P. lentiscus* extracts and the ways they affect the behavior of larvae and adults of *L. botrana*, in order to optimize their effectiveness as insect population control agents in vineyards.

**Supplementary Materials:** The following are available online at https://www.mdpi.com/article/10.3390/agronomy12040755/s1, Figure S1: PLFHe2 NMR, Figure S2: PLFHe2 GC-MS, Figure S3: Oleic acid, Figure S4: Linoleic Acid, Figure S5: Palmitic Acid, Figure S6: Stearic Acid, Figure S7: Fraction PLFHe4, γ-Sitosterol, Table S1: Tentative identification of constituents of Pistacia lentiscus hexane extracts by GC-MS.

**Author Contributions:** I.D. carried out the analytical protocols (NMR and GC-MS); I.D. and P.-C.B. carried out the insect rearing and the bioassays; M.K. supervised the entomological and chemical procedures and, with D.R., conceived and designed the experiments. M.K., D.R. and I.D. contributed to the writing of the paper. All authors have read and agreed to the published version of the manuscript.

**Funding:** This work received partial support from the project "An Open-Access Research Infrastructure of Chemical Biology and Target-Based Screening Technologies for Human and Animal Health, Agriculture and the Environment (OPENSCREEN-GR)" (MIS 5002691), which is implemented under the Action "Reinforcement of the Research and Innovation Infrastructure", funded by the Operational Programme "Competitiveness, Entrepreneurship and Innovation" (NSRF 2014–2020), and co-financed by Greece and the European Union (European Regional Development Fund).

**Institutional Review Board Statement:** Not applicable.

**Informed Consent Statement:** Not applicable.

**Data Availability Statement:** Not applicable.

**Conflicts of Interest:** The authors declare no conflict of interest.

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
