# Peer review of "Insecticidal Effect of Pistacia lentiscus (Anacardiaceae) Metabolites against Lobesia botrana (Lepidoptera: Tortricidae)"

_agronomy, doi:10.3390/agronomy12040755_

Round 1
Reviewer 1 Report
The Manuscript entitled ( Insecticidal effect of Pistacia lentiscus metabolites against Lobesia botrana) introduced good contribution for controlling of an important insect. The MS needs these modifications:
- Structure of most sentences need to rewrite specially long sentences need to separate it to 2 sentences such as lines (14-17), lines (26-29).
- Line 93: explain what are these ratios, according to dry weight ?
- Line 109: what is 300 ?
- Section of 2.2: most of this section are results. Please separate these results from methods to be in the results section
- Line 186: 5th instar larvae: Write as (5th larval instar) and the same in all MS.
- Line 240: (data not shown). Why ?. Please add figure of HPLC analysis and Table to show exact compounds.
- In all results, do not use P= 0.000, But use it as P≤ 0.000.
- In Table 1 and other tables and Figures as figure 3, use (a, letter of significance) with the biggest numbers.
- Lines 330-333: transfer this to Introduction in suitable position
- Lines 347-353: this is not adequate. Use literature on the same plant extract of this study or the same compounds. Otherwise, on the same pest. BUT, here, it is not same extract, fractions, or insect.
- In any organism (plant, insect), please add family and order if mentioned for the first time such as in line 371,372
- In general, discussion is poor, needs more information about these compounds and their effect on different insect pests especially plants from the same family and pest from the same order of the studied insect
Author Response
Dear Editor,
Please find in attachment the revised manuscript entitled :
" Insecticidal effect of Pistacia lentiscus (Sapindales: Anacardiaceae) metabolites against Lobesia botrana (Lepidoptera: Tortricidae) " Agronomy 11-00599 Version 1.
All the comments of the reviewers and the academic editor have been taken into consideration and we believe that the revised manuscript gives satisfactory answers to all their points. Especially we would like to thank them for their assiduous revision and helpful comments on the manuscript.
Specific comments requiring explanation are listed below. Page and Line numbers refer to the original pagination.
Academic editor comments
The suggestion have been adopted
Reviewer 1
Structure of most sentences need to rewrite specially long sentences need to separate it to 2 sentences such as lines (14-17), lines (26-29). CORRECTED
Line 93: explain what are these ratios, according to dry weight ? DONE
Line 109: what is 300 ? CORRECTED
Section of 2.2: most of this section are results. Please separate these results from methods to be in the results section. DONE Certain parts of the text were moved to the results section.
Line 186: 5th instar larvae: Write as (5th larval instar) and the same in all MS. DONE
Line 240: (data not shown). Why ?. Please add figure of HPLC analysis and Table to show exact compounds. DONE Table and chromatograph added as supplementary info so not to distruct the reader from the main results which deal with the mortality of larvae
In all results, do not use P= 0.000, But use it as P≤ 0.000. CORRECTED
In Table 1 and other tables and Figures as figure 3, use (a, letter of significance) with the biggest numbers. This comment is not clear to us. If it regards font changing we can do it along with other changes in the format of the MS that may be required
Lines 330-333: transfer this to Introduction in suitable position DONE
Lines 347-353: this is not adequate. Use literature on the same plant extract of this study or the same compounds. Otherwise, on the same pest. BUT, here, it is not same extract, fractions, or insect. CORRECTED
In any organism (plant, insect), please add family and order if mentioned for the first time such as in line 371,372 DONE
In general, discussion is poor, needs more information about these compounds and their effect on different insect pests especially plants from the same family and pest from the same order of the studied insect CORRECTED (expanded)
Reviewer 2 Report
The authors report the insecticidal activity of two fatty acids (Oleic and linoleic acids) extracted from Pistacia lentiscus. They demonstrated that these compounds showed larvicidal activity against larvae of Lobesia botrana in a dose-dependent manner. This finding is quite impressive and I have no doubt about it will interest most of the readers of this journal.
However, some flaws that I have noted have to be addressed.
In their title, I suggest the authors accompany the scientific name of their model insect, the Order and the Family in which their insect is classified.
In the statistical section (Lines 220-224), after running the ANOVA test, the authors used Tukey's studentized range honestly significant difference (HSD) to separate the means? The questions are:
Did they check for the normality and the homoscedasticity assumptions before running these tests?
What guided their choice of the HSD posthoc test?
I ask this because a lot of erroneous mean separations appear on the different tables of the manuscript. For instance,
In table 1, in the data obtained 24 hours after residual exposure; how do the authors explain that 4±1.5 and 15.5±3.3; 15.5±3.3 and 25±3.2 are not significantly different (they are associated with the same letter "a", "b" respectively)? Same question in table 3; data obtained with doses 160 and 200ug, 3hrs after topical application: 28±4 and 40±5.5 (accompanied by the same letter "c"). Doses 20 and 40ug, 24 hours after topical application: 6.7±3.8 and 20±6.3 (accompanied by the same letter "a"). Dose 40 and 80ug, 48hr after topical application: 22±4.7 and 44±5.1 (accompanied by the same letter "b"). At 72 hours after topical application, how can they mention that 22.7±5.4 and 22.6±5.2 are not significantly different to 44±5.1 (they are associated with the same letter "a")?
I suggest the author use an appropriate posthoc test for their data to correct this.
In lines 228, 240, the authors interpret data that are not shown. For the readers to fully believe them, I recommend the authors incorporate this data in the manuscript as supplementary results rather than simply not presenting them as it is the case.
In the legend of figure 3, I suggest the authors write "error bars of mean with the same letter within each column (fraction/h) are not significantly different".
Line 310, the authors wrote, " Whilst stearic and palmitic acid showed no activity........." I wonder where these results are shown in their manuscript. Please include this in your manuscript.
In the discussion section, I suggest the authors add 3 to 4 sentences describing the model of action (e.g. cytotoxicity) of fatty acids in insects. They can read the work of
Ren et al (2019) AW1 Neuronal Cell Cytotoxicity: The Mode of Action of Insecticidal Fatty Acids
Author Response
Dear Editor,
Please find in attachment the revised manuscript entitled :
" Insecticidal effect of Pistacia lentiscus (Sapindales: Anacardiaceae) metabolites against Lobesia botrana (Lepidoptera: Tortricidae) " Agronomy 11-00599 Version 1.
All the comments of the reviewers and the academic editor have been taken into consideration and we believe that the revised manuscript gives satisfactory answers to all their points. Especially we would like to thank them for their assiduous revision and helpful comments on the manuscript.
Specific comments requiring explanation are listed below. Page and Line numbers refer to the original pagination.
Academic editor comments
The suggestion have been adopted
Reviewer 2
In their title, I suggest the authors accompany the scientific name of their model insect, the Order and the Family in which their insect is classified. DONE
In the statistical section (Lines 220-224), after running the ANOVA test, the authors used Tukey's studentized range honestly significant difference (HSD) to separate the means? ……I suggest the author use an appropriate posthoc test for their data to correct this. CORRECTED (We used Duncun’s test)
In lines 228, 240, the authors interpret data that are not shown. For the readers to fully believe them, I recommend the authors incorporate this data in the manuscript as supplementary results rather than simply not presenting them as it is the case. CORRECTED (L 228, added “Specifically, the exhibited mortality was zero in all cases.” We believed we should not add one more table with all figures as “0”) (L 240 table and chromatograph added as supplementary info so not to distruct the reader from the main results which deal with the mortality of larvae).
In the legend of figure 3, I suggest the authors write "error bars of mean with the same letter within each column (fraction/h) are not significantly different". In this figure we compare the mean mortality of each fraction over the same time period.
Line 310, the authors wrote, " Whilst stearic and palmitic acid showed no activity........." I wonder where these results are shown in their manuscript. Please include this in your manuscript. CORRECTED Stearic and palmitic acid exhibited zero mortality and that is why we didn’t include it as a separete table.
In the discussion section, I suggest the authors add 3 to 4 sentences describing the model of action (e.g. cytotoxicity) of fatty acids in insects. DONE
We would also like to thank for your contribution to the preparation of this manuscript
Yours sincerely,
Dr. M.A. Konstantopoulou, on behalf of all co-authors of the manuscript
Round 2
Reviewer 1 Report
The revised manuscript has undergone an extensive revision. In my opinion, it can now be accepted if the editor agreed.